# Warranty Seal Deformation Identification for Product Warranty Violation

**DOI:** 10.3390/s22134688

**Published:** 2022-06-21

**Authors:** Md Rabiul Awal , Nurul Ain Zakaria, Muzammil Jusoh, Mahmoud A. Abdelghany , Muhammad Syarifuddin Yahya, Hidayatul Aini Zakaria

**Affiliations:** 1Faculty of Ocean Engineering, Technology and Informatics, Universiti Malaysia Terengganu (UMT), Kuala Nerus 21030, Terengganu, Malaysia; ainzakaria47@gmail.com (N.A.Z.); syarif_yahya@umt.edu.my (M.S.Y.); hidayatul@umt.edu.my (H.A.Z.); 2Faculty of Electronic Engineering Technology, Universiti Malaysia Perlis, Arau 02600, Perlis, Malaysia; muzammil@unimap.edu.my; 3Department of General Educational Development, Faculty of Science and Information Technology (FSIT), Daffodil International University, DIU Rd, Dhaka 1341, Bangladesh; 4Electrical Engineering Department, College of Engineering, Prince Sattam Bin Abdulaziz University, Wadi Addwasir 11991, Saudi Arabia; abdelghany@mu.edu.eg; 5Department of Electrical Engineering, Faculty of Engineering, Minia University, Minia 61519, Egypt

**Keywords:** product warranty violation, warranty seal, piezoelectric deformation, PVDF

## Abstract

Product warranty seals or stickers are criteria for after-sale warranty services. The unauthorized removal or modification of a seal will void the warranty. So far, there is no detection method to confirm the warranty, other than the visual inspection of the deformation of the seal. Hence, a system to detect, read, and record the ’warranty’ seal deformation is presented in this paper. A flexible piezoelectric sensor was used to determine the mechanical impacts of the seal. Three major impacts are discussed and evaluated in this paper—partial removal, complete removal, and drop deformations of the seal. These impacts were compared with the ambient responses to distinguish the conditions. All three impact cases show distinct characteristics in terms of sensor values, pulses, and pulse widths. For partial removal and complete removal of the seal, both cases exhibited maximum sensor values but differed in pulse and pulse width. A partially removed seal experienced the maximum number of pulses while complete removal experienced the maximum pulse width. However, if the seal experienced a drop impact, it showed lower sensor values, with the lowest pulse and pulse width. Hence, an algorithm was applied to generalize the conditions and decisions of warranty violations.

## 1. Introduction

Electronic gadgets are popular (even with the elderly generations) due to their compact geometries, lighter weights, and multilevel–multidimensional functions. Naturally, the production and distribution of these gadgets are the highest in the known time frames. The global consumer electronics market size was USD 729.11 billion in 2019, USD 689.45 billion in 2020, and is expected to be USD 989.37 billion in 2027, (Source: www.fortunebusinessinsights.com (accessed on 5 March 2022)). These numbers indicate the seriousness of the market. The distributions of these products mainly involve transportation facilities, whether direct to the customer or via a local distributor. The distribution is mainly done by cargo, via road, water, or air. Most electronic products are fragile by nature and can be easily harmed by unusual impacts during transportation. As such, these conditions are covered by product warranties. The warranty of a product confirms a period of service or the replacement of a product if damaged within reason. Usually, products contain warranty seals or patches as proof; hence, a broken seal signifies an unusual act, and the warranty is usually rejected in this case. Moreover, a product can be damaged without breaking the warranty seal. For instance, product damage can occur from water or fire impacts, which are easily detectable. In contrast, a poorly packed product can be damaged by uncontrolled transportation impacts, which are difficult to detect. Again, inexperienced or unauthorized exposure of sensitive components can lead to severe disorientation of a product. Hence, it is important for the manufacturing industry to detect the reason for failure—whether it is within the warranty violation conditions or not.

The impacts from transportation or illegal exposure can be classified as either strain or stress impacts and can be collected using a vibration sensor, i.e., a piezoelectric sensor. This sensor can detect the real-time strain and stress impacts experienced by the product. As such, it is expected that the sensor can detect the impacts of the different types of pressure conditions. Hence, whether the warranty seal is pulled off intentionally or damaged from other impacts, can be identified using this sensor. As such, relevant deformation patterns will be considered and can further be evaluated.

There are existing sensors for multipurpose functions [1,2,3,4]. More specifically, temperature sensor [5,6], humid sensor [7,8], current sensor [9], strain sensor [10,11], and more. However, these sensors are expensive and it is rather unnecessary to have a sensor that costs more than the package itself. Moreover, most of the time, they are difficult to use due to their multi-functional characteristics. Hence, a simple sensor is required, at a reasonable cost, and with ease of detection.

Thus, we propose a piezoelectric sensor-based warranty violation detection system. A flexible piezoelectric sensor was attached (as the warranty seal) to a package to be transported or in use. A system to detect, read, and record the piezoelectric deformations was designed using Arduino Uno. The data were stored on a memory card for further investigation—to find the deformation types. These types can be compared with ambient or natural deformations of the seal. The experimental design, results, and discussions are provided in the following sections.

## 2. Working Method

A flexible piezoelectric sensor has a wide range of responses. Typically, the responses vary from 0 to 5 V and higher, based on the product definition as well as the piezoelectric material volume. However, these responses can be detected and stored by a monitoring platform. These platforms can be permanent (system on chip) or temporary. We selected the Arduino-based temporary monitoring system due to its flexibility. However, the challenge with Arduino is that it cannot detect the piezoelectric responses in terms of voltages. Rather, it exhibits the responses in terms of sensor values. The highest value indicates the maximum voltage produced by the sensor. These sensor values can be adjusted. We distributed the total sensor values into 1024 units for our system. That means the highest piezoelectric effect will exhibit 1024 units as the maximum sensor value. The fluctuations in these values will identify the patterns for the aforementioned conditions, to verify the possible warranty violation.

## 3. System Design

The proposed system has three basic component blocks. The first component is the flexible piezoelectric sensor. We used an LDT0-028K piezoelectric sensor with a silver ink electrode, no mass version. The sensor was composed of 28-µm thick piezoelectric PVDF polymer film with screen-printed silver ink electrodes, laminated to a 0.125 mm polyester substrate, and fitted with two crimped contacts. The sensor had a 30 mm packaging length and a 13.208 mm packaging width. However, the active piezo layer was 23.368 mm ×10.16 mm ×28µm. The sensor could be operated from 0 °C to 85 °C. The sensor’s baseline sensitivity was 50 mV/g and its resonant frequency was 1.4 V/g at 180 Hz. The details of the sensor are depicted in Figure 1. As the sensor is flexible by nature, it can easily exhibit the mechanical properties of a warranty seal. The second block has an Arduino Uno module with a power supply. The third block holds the data memory. The total system set is available in Figure 2a.

### 3.1. Cases

Product damage can be classified into several conditions. Some damages are visibly determinable, for example, short circuits caused by conducting liquids or fire, broken, or heavily tampered. Some damages, on the other hand, are not visually detectable. For example, damages during transportation and intentional or unintentional modifications of the product components. Other than the natural causes, most of the aforementioned conditions indicate the warranty violation of the product and, evidently, the deformation of the warranty seal. These conditions can further be distributed in some cases based on the seal deformation. As such, four cases can be considered. Firstly, the seal is not deformed at all; secondly, intentionally or unintentionally deformed; and lastly, deformed due to the package displacements (dropped). Hence, (a) ambient responses, (b) partial applied pressure responses, (c) complete applied pressure responses, and (d) package drop test responses are the impacts required to identify the warranty violation. All the cases are depicted in Figure 2.

#### 3.1.1. Ambient Responses

In this case, the warranty seal is not deformed from any impacts other than the ambient responses. The piezoelectric sensor is left without any applied pressure. Thus, the sensor receives only the ambient response from the surroundings. Very low responses are expected from this case, as the sensor pressure will be minimum. The responses were recorded for 1 min 39 s and are illustrated in Figure 2a.

#### 3.1.2. Partial Removal Pressure Responses

In this case, an unintentional pressure is applied. Hence, it can be defined if the user mistakenly pulls off the warranty seal and restores it to its original spot. In our experiment, we considered that the seal would be partially pulled off and restored back. This case receives significant pressure deformations. The responses were recorded for 1 min 39 s. The setup can be found in Figure 2b.

#### 3.1.3. Complete Removal Pressure Responses

This case investigates the impact of unauthorized displacements of the warranty seal, i.e., the seal is removed completely. This case considers two types of impact responses—slow and fast responses. For fast displacements, the warranty seal is displaced completely in a quick manner. However, the seal is removed relatively slowly for the other case. Distinguishable responses are expected from this case. The responses were recorded for 1 min 39 s for both cases. Figure 2c demonstrates the experiment.

#### 3.1.4. Drop Deformation Test Responses

The possible displacements that occur during transportation are considered in this case. A transportable package was dropped on a hard surface several times to achieve harsh deformation impacts. This case is true if the product is damaged by the drop impact from the user. These impacts are expected to express the drop test responses. The responses were recorded for 1 min and 20 s. This case is pictured in Figure 2d.

#### 3.1.5. Definition

The corresponding voltage presented in Section 4 is not a real-time measured voltage. As we mentioned before, Arduino Uno cannot present the sensor output in volts. Hence, we measured the drawn source voltage and distributed it according to the sensor value range. The sensor value has a range of 1–1024, and the measured voltage is 2 volts. Therefore, each sensor value represents (2000/1024) = 1.953 mV; the 1024 sensor value corresponds to 2 V.

## 4. Discussions

The pressure responses from the four cases are presented in Figure 3. Figure 3a represents the ambient responses. The responses of partial seal removal pressure are expressed in Figure 3b, whereas, complete seal removal pressure responses are presented in Figure 3c,d for the quick and slow effects. Figure 3e shows the drop deformation responses. From the ambient responses in Figure 3a, we can see that the highest received response is 237 sensor values, with several picks of 230 and 219. They correspond to 462, 449, and 441 mV as generated voltages. The values achieved from the ambient conditions will be used for comparison with the rest of the cases. To do so, a threshold value will be settled based on the minimum sensor value achieved due to the deformation.

If any pressure is applied unintentionally, the condition can be defined as a partial deformation of the sensor. The found sensor values are at 1023 maximum, with several picks in this case. In addition, there is one pick value with 1020. They translate to generated voltages of 1998 and 1992 mV. The fluctuations can be seen in Figure 3b. For the case involving complete deformation of the sensor seal, the first condition was to apply a slow pressure. The responses are 1962, 1998, and 1935 mV in terms of voltages; the sensor values are 1005, 991, and 1023, respectively, to the generated voltages. In contrast, the quick pressure results are 1023 and 858 sensor values with 1998 mV as the highest, as pictured in Figure 3c,d.

An interesting scenario was found during the drop test, depicted in Figure 3e. Only one distinguishable sensor value was found as 511 with 998 mV of voltage. However, the closest pulse was found to be at a 413 sensor value with an 806 mV voltage response. These mentioned patterns can direct the differences among the cases and can be used to identify them. Hence, it is possible to settle the threshold value for the comparison between the ambient responses and the rest of the cases. The minimum sensor value for any found deformation is 511 (which is from the drop responses), while the maximum value for ambient response is 237. Hence, we can take any value less than 500 but above 300 as the threshold sensor value. We settled the threshold at 500 sensor values.

## 5. Decisions

We can see from the aforementioned results that a maximum sensor value of 237 indicates the ambient conditions, as in Figure 3a. Hence, a faulty device with this range of sensor values is more likely damaged by some other impact (or pre-damaged). It is recommended to trace the visible impacts (short circuits by the current overflow, water, or fire).

If a sensor value with 511 is found with the pattern of a single pulse (pulse width 1 s over the threshold value), then the device might have experienced a drop impact. If the damage is found immediately after delivery, there is strong reason to believe that the transportation caused the damage, otherwise, it might be the user who ’influenced’ the deformation. The scenario is presented in Figure 3e.

Two pulses with a maximum sensor value of 1023 in Figure 3c,d (pulse width 4 s over the threshold value) suggest to us that the device package might have had unauthorized exposure. Therefore, the warranty might be void.

The trickiest one is with a maximum sensor value of 1023, with several sharp pulses (in Figure 3b). The case condition suggests that the warranty seal is deformed or displaced partially. Hence, the seal was most likely displaced unintentionally and quickly restored. However, the impacts from the completely removed seal can have similar impacts if the seal is restored later on, as both cases can exhibit maximum sensor values (in Figure 3c,d). The average sensor value for complete removal is 186.33 (quick) and 261.92 (slow) whereas the partial removal impacts only have 94.7 sensor values. Hence, the cases are distinguishable based on the average sensor values. The details are presented in Table 1.

## 6. Algorithm

It is now possible to introduce an algorithm to define the warranty violation cases based on the aforementioned results and data summary from Table 1. The cases are differentiable based on the maximum sensor value, counted pulses over a threshold value, and pulse width (in seconds). Algorithm 1 presents the countable differences among the four cases.
**Algorithm 1 **Warranty violation algorithmWarranty Violation (Threshold Response, Max Response)Sensor Values, ⌊0⌋→⌈1023⌉Received Response, $R_{R}$ = ⌊0⌋→⌈SensorValues⌉;Max Response, $R_{max}$ = Max [$R_{R1},R_{R2},R_{R3},......R_{Rn}$]; where n = # of casesThreshold Response, $T_{R}=\exists !\left|\lceil R_{max}\rceil \right|$;Pulse over Threshold, $R_{S}$ = count [$R_{max}>T_{R}$]; ⌊1⌋→$\left\lceil R_{S}\right\rceil$;Pulse Width, $P_{w}$ = duration [$R_{max}>T_{R}$]; ⌊1⌋→$\left\lceil P_{w}\right\rceil$;**while** 
$R_{max}\geq T_{R}$ 
**do**     Warranty Violation = POSITIVE     **if** $R_{max}\geq T_{R}$ && $R_{S}=\lfloor R_{S}\rfloor$ && $P_{w}=\lfloor P_{w}\rfloor$ **then**            STATUS = Warranty Violation [Dropped Seal Deformation]     **else if** $R_{max}\geq T_{R}$ && $R_{S}=\lceil R_{S}\rceil$ && $\left\lfloor P_{w}\rfloor <P_{w}<\lceil P_{w}\right\rceil$ **then**            STATUS = Warranty Violation [Partial Seal Deformation]     **else if** $R_{max}\geq T_{R}$ && $\left\lfloor R_{S}\rfloor <R_{S}<\lceil R_{S}\right\rceil$ && $P_{w}=\lceil P_{w}\rceil$ **then**            STATUS = Warranty Violation [Complete Seal Deformation]     **end if****end while****if** 
$T_{R}>R_{max}$ **then**     STATUS = Warranty Violation Otherwise**else**     STATUS = ERROR**end if**

## 7. Conclusions

This paper presents an approach to detect unauthorized warranty violations. Thus, a simple flexible piezoelectric sensor was used in the Arduino Uno platform to represent the mechanical impacts of a warranty seal. These sensor impacts illustrate the fluctuations from the seal under different types of pressure. As such, four cases were considered in this work to verify the eligibility of a warranty. As (a) ambient responses, (b) partially applied pressure responses, (c) completely applied pressure responses, and (d) package drop responses. We can see that the sensor exhibits a maximum of 237 and an average of 173.01 sensor values for ambient conditions. However, when the sensor is displaced partially, it shows a maximum of 1023 sensor values with several sharp pulses (maximum pulse width 2 seconds). It also returns a similar impact when the seal is replaced. Nevertheless, for complete displacements of the sensor, the counted pulses are dropped to 2, while the maximum pulse width is increased to 4 seconds and the maximum sensor values remain the same as the partial displacements. When the sensor experiences the drop deformation, it shows a significantly lower sensor value of 511 with an average value of 210.12 and a maximum pulse width of 1 seconds. From these analyzed results, we can see that the deformation type of a warranty seal can be characterized. By following the types, the impacts on the warranty seal can be identified. Thus, the unauthorized exposure of the product components can be detected and further steps can be taken according to the warranty protocol. 

## Figures and Tables

**Figure 1 sensors-22-04688-f001:**
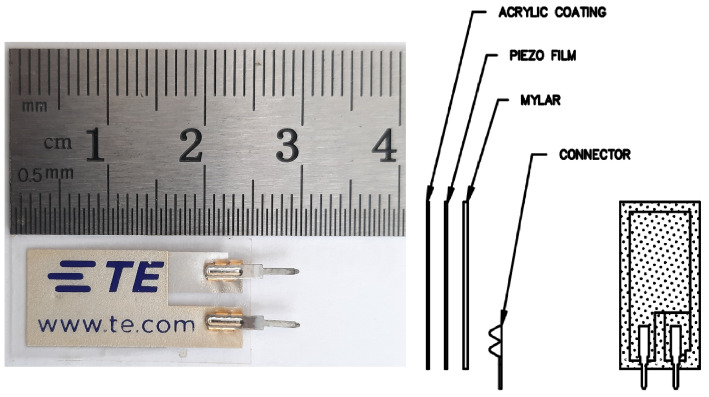
Sensor descriptions. Sensor illustrations (**left**), sensor dimensions (**right**).

**Figure 2 sensors-22-04688-f002:**
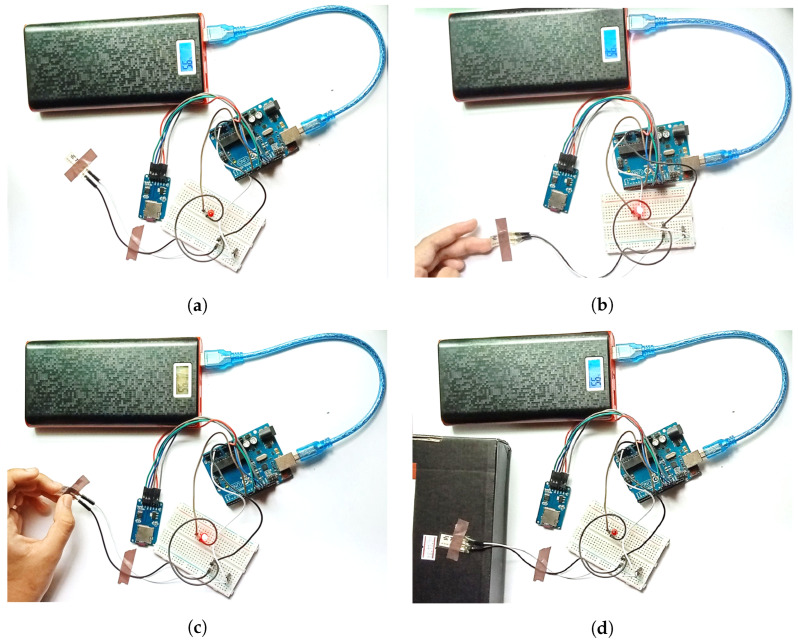
Experimental Set ups. (**a**) Ambient responses test; (**b**) Partially removed sensor; (**c**) Completely removed sensor; (**d**) Drop deformation test.

**Figure 3 sensors-22-04688-f003:**
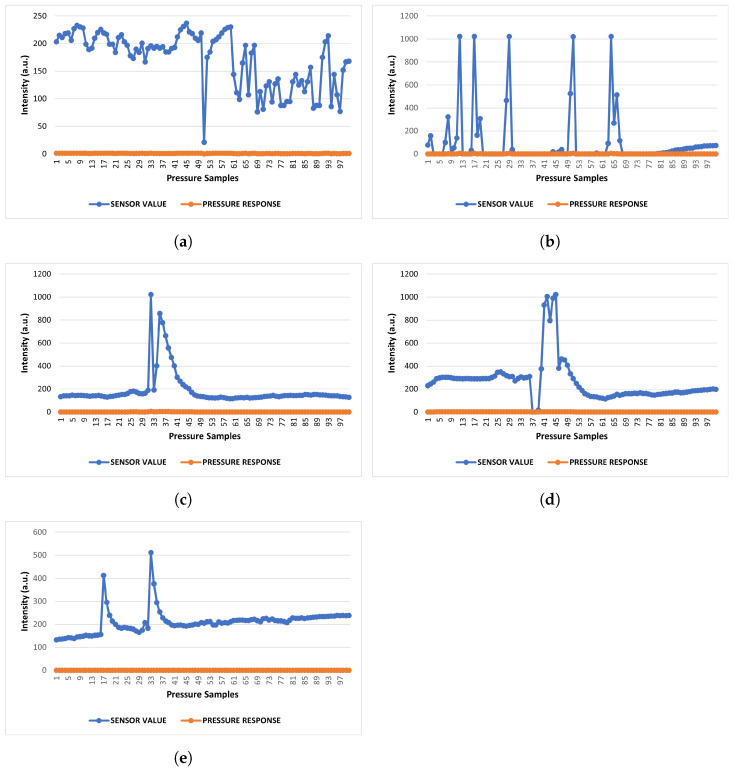
Pressure responses from different cases. (**a**) Ambient responses; (**b**) Partial removal pressure responses; (**c**) Complete removal pressure responses (quick); (**d**) Complete removal pressure responses (slow); (**e**) Drop deformation test responses.

**Table 1 sensors-22-04688-t001:** Summary of the results.

Cases	Max Sensor Values	Avg. Sensor Threshold	Pulse over Threshold	Max Pulse Width over Threshold
Ambient	237	173.01	None	None
Partial Removal	1023	94.7	6	2 s
Complete Removal	1023	186.33 (quick)	2	4 s
		261.92 (slow)		
Dropped	511	210.12	1	1 s

## Data Availability

Not applicable.

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
