# Peer review of "Warranty Seal Deformation Identification for Product Warranty Violation"

_sensors, 2022, doi:10.3390/s22134688_

Round 1
Reviewer 1 Report
Comments:
The paper deals with the use of commercial piezoelectric sensors to detect
unauthorized warranty violation.
1) Is there a criterium for the choice of the time to record sensors answer?
2) Why the authors choose two different record times according to the
experimental condition to detect sensor response?
3) What about the repeatability of the sensors responses? Have you performed
more than one measure for each investigated experimental condition?
4) Is it possible to define a sort of sensitivity of the sensors?
Others:
1) English should be improved, there are some mistakes (i.e. line 76, line 164,
etc)
2) Why in the keywords the authors mention “ZnO”? The used sensors are
based on PVDF polymer piezoelectric material…
Author Response
Point 1: Is there a criterium for the choice of the time to record sensors answer?
Response 1): As per our knowledge and existing literature, we did not come across any criteria for the recording time. The main scope of the manuscript is to detect the unusual responses or deformations in the sensor. These unusual responses or deformations happen only in a short period of time. Thus, we need those deformations to detect warranty violations. We have considered a larger time frame which results in unnecessary void data. However, this reduced time frame covers the important data span. Hence, we have focused only on the analytical importance of the samples.
2) Why the authors choose two different record times according to the experimental condition to detect sensor response?
Response 2): Two different record times (1 min 39 seconds and 1 min 41 seconds) were used to keep the same sample size which is 100 for all the cases. For each reading, we have collected the exact same number of samples for the sack of clear comparison availability. So, we need to compare all the cases with the ambient response as the reference case. Thus, we would be able to compare the ambient response with the rest of the cases.
3) What about the repeatability of the sensors responses? Have you performed more than one measure for each investigated experimental condition?
Response 3): Yes, the experiment has a total of 5 different set up to cover different cases. For each case, several readings were carried out for the analyses.
4) Is it possible to define a sort of sensitivity of the sensors?
Response 4): We have described the sensor sensitivity in section 2 (Working method). Sensor sensitivity can be defined as sensor response against the applied pressure. Sensor output can be obtained as volt. However, Arduino can not collect the sensor response in volts. Hence, due to the recording limitations in the Arduino platform, we have taken the relative value as Intensity (a.u.) vs pressure sample. The sensor response is distributed to a maximum of 1024 units, hence, it exhibits 1024 Intensity (a.u.) when maximum pressure is applied. As such, 0 when no pressure is available. We have taken 100 samples for the pressure applied and the related responses for each case. Sensor sensitivity definition in section 2 (Working method) is attached here for reference:
“Flexible piezoelectric sensor has a wide range of responses. Typically, the responses varied from 0 V to 5 V and higher, based on the product definition, as well as the piezoelectric material volumes in it. However, these responses can be detected and stored by a monitoring platform. These platforms can be permanent (system on chip) or temporary. We have selected the Arduino based temporary monitoring system due to its flexibility to use. The challenge with the Arduino however is, that it can not detect the piezoelectric responses in terms of voltage. Rather, it exhibits the responses in terms of sensor values. The highest value indicates the maximum voltage produced by the sensor. These sensor values can be adjusted. We have distributed the total sensor values into 1024 units for our system. That means the highest piezoelectric effect will exhibit 1024 units as the maximum sensor value. The fluctuation in these values will identify the pattern for the aforementioned conditions to verify the possible warranty violation.”
Others:
1) English should be improved, there are some mistakes (i.e. line 76, line 164,
etc)
Response 1): The manuscript is revised as instructed by the anonymous reviewer. Moreover, we have done grammatical checking throughout the manuscript once again.
2) Why in the keywords the authors mention “ZnO”? The used sensors are
based on PVDF polymer piezoelectric material…
Response 2): Really sorry for our typing mistake. The keyword is corrected to PVDF from ZnO, ZnOas suggested by the reviewer. Thank you for the constructive comments.

Reviewer 2 Report
I don't think the manuscript is suitable to publish in this version due to lack of the innovation.
1. Please address some highlights in this manuscript. 2. Although the piezoelectric sensor is bought from some company, please introduce it in detail. 3. In Fig. 3, how about the results of repeated measurements under different conditions, especially, with different forces or angles? Please comment the conformance testing. 4. How about the effects of temperature, humidity, vibration and shock on the pressure responses of the sensor? 5. The coordinate axes should have titles and units in Fig. 3.Author Response
Point 1: Please address some highlights in this manuscript.
Response 1: Necessary key points are included in the Abstract and Conclusions. As follows,
“Product warranty seal or sticker is one of the criteria for after sale warranty services. Unauthorized removal or modification of this seal will void the warranty. So far, there is no detection method other than the visual inspection of the seal deformation to confirm the warranty. Hence, a system to detect, read and record the warranty seal deformations is presented in this paper. A flexible piezoelectric sensor is used to determine the mechanical impacts on the seal. Three major impacts are discussed and evaluated in this paper. Namely, partial removal, complete removal and drop deformations of the seal. These impacts are then compared with the ambient responses to distinguish the conditions. All the three impact cases show distinct characteristics in terms of sensor values, pulse, and pulse width. For partial removal and complete removal of the seal, both cases exhibits maximum sensor values but differs in pulse and pulse width. Partially removed seal experience maximum number of pulses while complete removal has the maximum pulse width. However, if the seal experience drop impacts, it shows lower sensor values with lowest pulse, and pulse width. Hence, an algorithm is applied to generalize the conditions and decisions of warranty violations.”
“An approach to detect unauthorized warranty violation is presented in this paper. Thus, a simple flexible piezoelectric sensor is used in the Arduino Uno platform to represent the mechanical impacts of a warranty seal. These sensor impacts can illustrate the fluctuations from the seal under different types of pressure. As such, four cases are considered in this work to verify the eligibility of warranty. As, (a) ambient responses (b) partially applied pressure responses (c) completely applied pressure responses and (d) package drop responses. We can see, the sensor exhibits maximum 237 and average 173.01 sensor values for ambient condition. However, when the sensor is displaced partially, it shows maximum 1023 sensor values with several sharp pulses (maximum pulse width 2 seconds). It also returns the similar impact when the seal is replaced back. Nevertheless, for complete displacements of the sensor, the counted pulses are dropped to 2, while maximum pulse width is increased to 4 seconds and maximum sensor values remain same as partial displacements. When the sensor experience the drop deformation, it shows significantly lower sensor value of 511 with average value of 210.12 and maximum pulse width of 1 second. From these analyzed results, we can see that, the deformation type of a warranty seal can be characterized. By following the types, the impacts on the warranty seal can be identified. Thus, the unauthorized exposure of product components can be detected and further steps can be taken according to the warranty protocol. ”
- Although the piezoelectric sensor is bought from some company, please introduce it in detail.
Response 2: We have added an introduction of the sensor in Section 3 along with the picture descriptions as follows,
“The first component is the flexible piezoelectric sensor. We have used a LDT0-028K Piezoelectric Sensor with silver ink electrode, mass less Version. The sensor is comprising a 28 µm thick piezoelectric PVDF polymer film with screen printed silver ink electrodes, laminated to a 0.125 mm polyester substrate and fitted with two crimped contacts. The sensor has a 30 mm packaging length with a 13.208 mm packaging width. However, the active piezo layer has 23.368 mm ×10.16 mm ×28 µm. The sensor can be operated from 0â—¦C to 85â—¦C temperature. Sensors baseline sensitivity is 50 mV/g while it is 1.4 V/g at 180 Hz resonant frequency. The details of the sensor are depicted in Figure 1.”
- In Fig. 3, how about the results of repeated measurements under different conditions, especially, with different forces or angles? Please comment the conformance testing.
Response 3: The experiment has a total 4 different set up and each set up was investigated with multiple readings. The set up are for Ambient responses, Partial removal pressure responses, Complete removal pressure responses and Drop deformation test responses. The experimental setups are available in figure 2. We did multiple experiments using this set up to collect multiple readings. Multiple readings confirm the same pattern as we have presented in the manuscript.
- How about the effects of temperature, humidity, vibration and shock on the pressure responses of the sensor?
Response 4: We did perform repeated experiments at room temperature. The main objective for choosing the room temperature is that most of the transportation or household use happens at this temperature. The operating temperature of the sensor ranged from 0oC to 85oC. However, the study of the effect of temperature and humidity is out of the scope of this work. Nevertheless, the impact of the vibration and shock is the main scope of this paper which are described in detail in the manuscript.
- The coordinate axes should have titles and units in Fig. 3.
Response 5: Thanks very much for the suggestion. We have corrected the figure title and added the units as suggested. A sample snapshot is given below.

Reviewer 3 Report
Interesting paper on seal breaking sensors based upon mechanical strain.
Page 2, what are the advantages of this kind of sensor over a metallic particle/polymer strip which when deformed would change its geometry and hence electrical resistance.
Page 4, are there a sufficient number of data analyzed here to support the main conclusions regarding the discrimination between different stimuli.
Page 7, could a second sensor for temperature, continuity or water vapor be added to confirm that the seal is broken.
Author Response
Point 1: Page 2, what are the advantages of this kind of sensor over a metallic particle/polymer strip which when deformed would change its geometry and hence electrical resistance.
Response 1: The major advantage of this kind of sensor is that it is the sensor deformation we need. The sensor offers detectable deformations when pressed only. From the results and discussions, we can see that the ambient responses do not have a particular pattern. In contrast, all the deformed cases exhibit patterned responses (which are distinguishable of course). Hence, the patterned responses are the focus of this study.
Page 4, are there a sufficient number of data analyzed here to support the main conclusions regarding the discrimination between different stimuli.
Response 2: The experiment has a total 5 different set up and each set up was investigated with multiple readings. The readings reflect the same patterns.
Page 7, could a second sensor for temperature, continuity or water vapor be added to confirm that the seal is broken.
Response 3: We appreciate the concern of the reviewer. We have mentioned in the introductions that the experiments will investigate the impacts of poorly packed products and inexperienced or unauthorized exposures. The broken seal, however, is expected to be visually detectable. Thus, it is out of the scope of our work. We have mentioned in the Introductions as follows,
“Besides these obvious observations, a product can be damaged without breaking the warranty seal. For instance, product damage can occur from water or fire impacts which are easily detectable. In contrast, the poorly packed product can be damaged by the uncontrolled transportation impacts which are difficult to detect. Again, inexperienced or unauthorized exposure of the sensitive components can lead to the severe disorientation of the product, to out of order as well. Hence, it is important for the manufacturing industry to detect the reason of failure whether within the warranty violation conditions or not.”

Round 2
Reviewer 2 Report
I don't think that the question of "How about the effects of temperature, humidity, vibration and shock on the pressure responses of the sensor?” is well answered. Please reconsider it.
Author Response
Response to Reviewer 2 Comments
Please see the attachment.

Reviewer 3 Report
Paper is improved with your corrections
Author Response
Response to Reviewer 3 Comments
Point 1: Moderate English changes required.
Response 1: We proofread the entire manuscript as suggested by the reviewer. We thank the reviewer for the constructive comments.
